# The Role of Oxytocin in Abnormal Brain Development: Effect on Glial Cells and Neuroinflammation

**DOI:** 10.3390/cells11233899

**Published:** 2022-12-02

**Authors:** Marit Knoop, Marie-Laure Possovre, Alice Jacquens, Alexandre Charlet, Olivier Baud, Pascal Darbon

**Affiliations:** 1Laboratory of Child Growth and Development, University of Geneva, 1205 Geneva, Switzerland; 2Faculty of Medicine, Université Paris Cité, Inserm, NeuroDiderot, 75019 Paris, France; 3Centre National de la Recherche Scientifique, Institute of Cellular and Integrative Neuroscience, University of Strasbourg, INCI UPR3212, 67000 Strasbourg, France; 4Division of Neonatology and Pediatric Intensive Care, Children’s University Hospital of Geneva, 1205 Geneva, Switzerland

**Keywords:** oxytocin, developing brain, neuroprotection, neuroinflammation, amygdala, microglia, astrocytes

## Abstract

The neonatal period is critical for brain development and determinant for long-term brain trajectory. Yet, this time concurs with a sensitivity and risk for numerous brain injuries following perinatal complications such as preterm birth. Brain injury in premature infants leads to a complex amalgam of primary destructive diseases and secondary maturational and trophic disturbances and, as a consequence, to long-term neurocognitive and behavioral problems. Neuroinflammation is an important common factor in these complications, which contributes to the adverse effects on brain development. Mediating this inflammatory response forms a key therapeutic target in protecting the vulnerable developing brain when complications arise. The neuropeptide oxytocin (OT) plays an important role in the perinatal period, and its importance for lactation and social bonding in early life are well-recognized. Yet, novel functions of OT for the developing brain are increasingly emerging. In particular, OT seems able to modulate glial activity in neuroinflammatory states, but the exact mechanisms underlying this connection are largely unknown. The current review provides an overview of the oxytocinergic system and its early life development across rodent and human. Moreover, we cover the most up-to-date understanding of the role of OT in neonatal brain development and the potential neuroprotective effects it holds when adverse neural events arise in association with neuroinflammation. A detailed assessment of the underlying mechanisms between OT treatment and astrocyte and microglia reactivity is given, as well as a focus on the amygdala, a brain region of crucial importance for socio-emotional behavior, particularly in infants born preterm.

## 1. Introduction

Every year, 30 million infants worldwide are born with intra-uterine growth restriction (IUGR) and 15 million are delivered preterm [1,2]. These two conditions are the leading causes of ante/perinatal stress and brain injury responsible for neurocognitive and behavioral disorders in more than 9 million children each year [3]. Possible developmental consequences of neonatal trauma include behavioral disorders such as autism spectrum disorder (ASD), learning disabilities and cerebral palsy [4,5,6]. A major factor that contributes to the progression of brain injury is neonatal systemic neuroinflammation, characterized by the activation of the immune system due to preterm birth itself, but also by hypoxemic events or postnatal pro-inflammatory complications that can arise [7,8]. This systemic inflammation is tightly linked to the activation of microglia and astrocytes in the developing brain [9]. Inflammatory insults to the developing brain lead to alterations in global brain tissue growth rates [10] as well as microstructural alterations in white matter networks including the associative and limbic cortico-basal ganglia-thalamocortical circuits, involving the dorsolateral prefrontal cortex, the orbitofrontal cortex, and the amygdala [11,12]. These structural brain changes have been linked with deficits in early emotion processing and emotion regulation, as well as attention, executive control and social reasoning disorders [13,14,15,16,17,18].

Further adding to the increased risk of adverse neural effects following perinatal complications is the subsequent admission of the newborn to the neonatal intensive care unit (NICU) [19]. The NICU is an environment that, though necessary and optimal for medical caregiving, can induce increased early life stress created by frequent and inappropriate sensory stimuli, invasive and painful procedures, and chronic early maternal separation [20]. A cohort study of 180 newborns showed that the degree of invasive procedures at the NICU was associated with changes in long-term network-specific maturational covariance patterns in the brain, which can render the infant more susceptible to stress-related psychopathology in later life [21].

Finding ways to protect the brain in the neonatal period, to stimulate brain development and to reduce the impact of early life challenges, is thus a key objective for developmental translational and clinical neuroscience. Environmental enrichment, personalized care, skin-to-skin contact and music exposure all appear to confer positive effects on brain structure and function in a newborn [22,23,24,25]. ‘Bedside’ data such as these, therefore, propose a neuro-stimulating effect of the neuropeptide oxytocin (OT), as all these interventions show to increase OT levels in the infant (and mother/father alike) [24,26]. Moreover, perinatal complications such as preterm birth have been shown to reduce OT release as part of an increased stress response [27], and the brain regions that are affected by perinatal complications largely show an enriched expression of OT receptors [28].

The amygdala is an important brain structure for the OT system, which is also specifically affected by preterm birth and implicated in behavioral disorders that have a perinatal origin [11]. The amygdala is a key region associated with the functioning of the limbic system, specifically with the processing of fearful stimuli, and its functions are known to be regulated by OT [29,30,31,32]. Consequently, preterm children can show difficulties with socio-emotional processing. Indeed, preterm-born children show decreased amygdala volumes and increased social anxiety at adult ages, compared to term-born neonates [33]. Moreover, a longitudinal functional MRI study on preterm-born children found that amygdala resting-state activity at term-equivalent age is associated with socio-emotional outcomes at 4 years of age [34]. These reports underline that the effects induced by perinatal complications such as preterm birth can have a long-term effect on structural and functional brain development. OT treatment seems to improve the functioning of these brain areas, including the amygdala [35]. The association of OT with neonatal brain protection is thus two-fold: the OT system appears affected in neonatal brains that suffered from perinatal complications but increased OT levels through interventions show a potential protective effect. Yet, how this potential neuroprotective effect occurs remains to elude. The underlying molecular mechanisms of how OT can offer a neuroprotective effect in a state when the neonatal brain is challenged in the form of injury and neuroinflammation are underexplored.

## 2. The Oxytocin System in the Brain

Research on OT was pioneered in the 1920s by German anatomist Ernst Scharrer, after he identified unusual, large-shaped “glandule-like” cells in the hypothalamus of fish [36]. A full anatomical, morphological and functional assessment would follow in the next 50 years, complemented by the Nobel-prize awarded for the synthesis of OT to Vincent du Vigneaud in 1955 [37].

### 2.1. Oxytocin Synthesis and Release to the Oxytocin Receptor

The human OT gene is located on chromosome 20p. It consists of 3 exons and encodes for Pre-Pro-OT-Neurophysin I. This OT prohormone is cleaved successively by different enzymes into intermediate OT forms, and finally the mature, amidated OT form. OT-prohormone (at E14.5) and intermediate OT forms (at E16.5) are already detected in the embryonic phase, but mature OT is only detected after birth [38,39]. Although immature OT forms are gaining support to play a function (yet unspecified) in disorders such as ASD, the main form of functional OT is mature OT.

In the brain, the neuropeptide OT is mainly produced in the hypothalamus. It is principally produced in magnocellular and parvocellular neurons of the periventricular nucleus (PVN) and supraoptic nucleus (SON), with some additional OT-producing neurons in the nucleus circularis and rostral supraoptic nucleus [40]. OT is spread through the central nervous system in multiple ways. First, via axonal release, which classically takes place at the synapse but also includes “en passant” release from axonal varicosities [41]. In this type of OT release, action potentials trigger OT release from axonal synapses or boutons that directly project to synapses in various brain regions [42]. Where synaptic OT release is quick, *en passant* release is slow and diffusion-based, causing a 60–90 s delay in response [43]. Secondly, OT neuropeptide is spread through the brain via somatodendritic release, where OT is stored and released locally in the PVN in large dense-core vesicles via exocytosis from dendrites or the cell soma [44,45,46]. This way, OT can exert autocrine effects on its own cell, thereby creating a strong self-regulatory mechanism. Somatodendritic OT release can also affect surrounding neurons and glia cells. This type of OT release is not targeted for synaptic terminals, but involves volume transmission, in which OT can travel long distances in the brain via passive diffusion, or by bulk flow via the extracellular fluid or cerebral spinal fluid (CSF) that is accessed through the third ventricle located near the PVN [44,47]. Volume OT transmission is made possible by the long half-life of central OT (about 20 min in CSF). Of note, the difference in release mechanisms make OT neurons capable of managing axonal and dendritic release independently from each other [48]. In parallel, OT can be released into the bloodstream via the posterior pituitary gland [49] and act as a hormone. These two pathways are independent. Indeed, plasma OT concentrations show no relationship with OT levels in the CSF [28]. However, some studies have identified simultaneous OT projections to regions of the forebrain as well as the posterior pituitary [30], showing that certain situations such as stress, can invoke an increase in both central and peripheral OT release [50].

The effects of OT are implemented via its binding to the oxytocin receptor (OTR), which is a seven-transmembrane G-protein-coupled receptor. Expression of OTR has been found on excitatory and inhibitory neurons throughout the brain (Human protein atlas: https://www.proteinatlas.org/ENSG00000180914-OXTR/brain), but also on astrocytes and its presence on microglia is debated (see Section 5). OTR is not exclusively bound by OT, as it can also be bound (yet with a much lower affinity) by vasopressin (AVP), a nonapeptide that is only 2 out of 9 amino-acids different from OT, and located on the same chromosomal locus [28,51]. Likewise, OT can bind to AVP receptor subtypes V1a, V1b and V2 [52]. However, OT binds to OTR with a much higher affinity than to AVP receptors: the receptor affinity (K_i_) of OT is 1.0 nм for OTR and 71 nм, 294 nм and 89 nм for V1a, V1b and V2, respectively [53]. Therefore, in this review, we will consider the actions of OT as acting exclusively via the OT receptor. For excellent papers on AVP and OT interaction, we refer to [40,44,54].

### 2.2. Development of the Oxytocin System from the Embryonic to Juvenile Age

OTR autoradiography assays, mRNA assessments and immunohistochemistry have been used to map the central OT system, which includes the main targets of OT projections, and the areas with high OTR expression [55,56,57,58,59]. The OT system shows a large spatial and temporal plasticity of OTR expression during development. Moreover, the development of the OTR system shows different trajectories between humans and rodents, but also between mice and rats [28,55,60]. To improve translational opportunities between OTR experiments in rodents to eventual human patients, it is needed to highlight the similarities and differences of the developmental OTR system. Figure 1 summarizes the development of OTR expression from embryo to juvenile stages in rats, mice and humans. For more information on the specific proteins involved in the OTR network, we refer to Chatterjee and colleagues [61] who created an extensive map of the OTR pathway based on data from 1803 screened articles.

#### 2.2.1. Oxytocin Receptor System of the Developing Rat Brain

Most of the studies on the OTR system have been performed in rats. Mature OT is produced from E21 onwards, but OT receptors are developed already before [62]. The earliest sighting of OTR mRNA in rats was found at E13/E15, in the posterior portion of the neuronal tube that will become the vagal motor nucleus [58]. At E20, OTR expression has spread to the tenia tecta (a component of the olfactory cortex), piriform cortex, caudate putamen, ventral tegmental area and several nuclei of the brainstem (Figure 1). It is only after P1 that OTR mRNA is starting to be expressed in the thalamic nuclei [58]. After birth, the distribution of OTR in rats becomes more localized and OTR expression starts to appear in new locations as well (Figure 1). The expression of OTR mRNA in the PVN peaks at P7 and remains stable throughout adulthood [28]. Around P10, a ‘pediatric’ pattern of OTR expression is found throughout the rat brain (Figure 1). In rats, mice, and mammals in general, no central OT projections are found during the embryonic and early postnatal phase, which indicates that early life OT signaling happens predominantly through dendritic release and volume transmission [60]. Axonal OT projections start emerging in the pediatric period. As such, axonal OT offers addressed modulation to the development of detailed skills that the young rodent undergoes in this period [60]. The distribution of OTR expression is highly transient during development, and only certain regions show OTR expression both in early life and in adulthood [63,64]. One of these regions is the amygdala, which shows clear OTR expression in all stages of rat development (Figure 1). Areas that show a surge in OTR expression that consequentially disappears after the postnatal period include the parietal and cingulate cortices, the caudate putamen and the PVN [38]. There are two periods in rat development that show a particular strong spatial shift in OTR expression. They are the third postnatal week (finalized at P18) and the juvenile age [28]. In juvenile rats, decreased expression of OTR can be found for the cingulate cortex, anterior thalamic nuclei, and ventral tegmental area (Figure 1). This suggests that these regions are involved in the juvenile-characteristic changes in social behavior such as an increase in risk-taking behavior and the self-regulatory system [65]. The rat adult OTR pattern is reached by P60.

#### 2.2.2. Oxytocin Receptor System of the Developing Mouse Brain

Initial OT mRNA expression in mice is found at E15.5, and by P0, OT neurons are found in almost all hypothalamic nuclei [38,66]. Unlike rats, the production of mature OT in mice does not start until after birth [67]. Mouse OTR mRNA can be detected as early as E12.5, and from E16.5 onwards, OTR expression is shown in ventricular, subventricular zones as well as in amygdala (Figure 1) [68]. There appears to be a sex-effect, in that females show much higher levels of OTR transcriptome in the embryonic and postnatal period compared to males [68]. OTR expression increases generally during the first two postnatal weeks, and pediatric OTR patterns are reached between P7 and P14. At this age, OTR expression is found throughout the brain, including the olfactory bulbs, neocortex, hypothalamus, hippocampus and amygdala (Figure 1) [59]. Between the pediatric and the juvenile age, the general mouse OTR expression patterns are replaced by more region-specific up-or downregulation of OTR expression [59]. Namely, OTR expression remains high until P21 in the lateral septum and dorsal CA3, but OTR expression in the neocortex peaks at P14. Similar to the rat, this developmental peak of receptor expression in the neocortex of the mouse coincides with major developments in synaptic pruning and wiring that are taking place, which suggests that OT is involved in these synaptic processes [59]. The juvenile OTR pattern in the mouse is similar to the adult pattern. However, the juvenile age shows a characteristically higher OTR expression in the dorsal/intermediate lateral septum, the cingulate cortex and the posterior PVN, compared to adult-mice [69]. Moreover, OTR expression at the juvenile age is increased in the ventromedial hypothalamus, a region that is involved in social behaviors that develop after puberty (sexual and aggressive behaviors) [70]. The pattern of increased OTR expression in the juvenile age is seen in mice and rats alike. Some differences between these species are that juvenile mice show an abundant yet transient distribution of OTR in the cingulate cortex and hippocampal regions, which is much lower, but chronic in juvenile rats [69]. The cingulate cortex is important for reward-based behavior, which suggests that the different OTR expression patterns in this region between rats and mice could attribute to the species-specific differences in social behavior [71,72]. For further visualization of postnatal OTR mapping in mouse coupes between P7 and P56 we refer to the online tool developed by the Kim lab (https://kimlab.io/brain-map/OTR/).

#### 2.2.3. Oxytocin Receptor System of the Developing Human Brain

Human brain development is different from rodents in that many key processes happen earlier on the developmental timeline, including in the prenatal phase [73,74] (see [75] for an extensive overview of temporal differences in brain development between rodents and humans). As such, the neurodevelopmental equivalent of the human brain at birth corresponds to the rat and mouse brain at P10 [76]. Moreover, the human OTR system already reaches its infant-like spatial pattern before birth, while in rats and mice, this does not happen before P10 and P7–P14, respectively [38].

OT is detected in human embryos as early as 11 weeks of gestational age (GA) [54]. The production and migration of OT neurons are completed around 25 weeks GA, but their morphological and electrophysiological maturation does not finalize until the first two weeks after birth [77]. Transcriptomic analysis of human brains shows a progressive increase in OTR mRNA expression in the embryonic period [56]. This included the amygdala, mediodorsal thalamus, hippocampus, striatum and numerous neocortical areas (Figure 1). In the third trimester, a peak in OTR expression is found for the striatum, hippocampus and orbitofrontal, dorsolateral prefrontal and caudal superior temporal cortices [57]. This is followed by OTR increases in the amygdala, hippocampus, primary visual cortex and inferolateral temporal cortex just before birth (Figure 1). The peak level of OTR expression in humans occurs during early childhood (6 years of age), which coincides with increased OTR levels in the mediodorsal nucleus of the thalamus and medial prefrontal cortex [57]. Later childhood phases (12 years of age) show an increase in OTR expression in the medial prefrontal and cerebellar cortices, as well as the mediodorsal hypothalamus (Figure 1). Compared to the pediatric phase, the juvenile stage of the human OTR system seems characterized by a strong decrease in mediodorsal thalamus OTR expression, a moderately lower expression in the medial prefrontal cortex, as well as increased OTR expression in the striatum and primary auditory and visual cortices (Figure 1) [57]. Similar to rats and mice, some regions in the human OTR system show a peaked expression in the pediatric phase, followed by a decrease during adolescence [55,56,57,58]. These regions have key functions for brain development in the pediatric phase. They include the cerebellar cortex and primary motor cortex (linked with the immense development of the motor system during infancy [78]), and the primary visual and somatosensory cortices (the somatosensory cortex similarly peaks at the pediatric age in rodents). OTR expression notably decreases in the juvenile age in the striatum and the medial prefrontal cortex, which are two regions important for the control of behavior and social reward processing. The adolescent period is characterized by an increase in risk-taking behavior and negative health outcomes, which is associated with increased reactivity of the striatum in response to rewards [79]. The fact that OTR expression is decreased in this region during the adolescent age fits in this line, for it means that the limbic system has decreased control over the striatal response to rewards.

Although there are different OTR patterns across brain development in mice, rats and humans, the development in the amygdala stands out because it is one of the few regions that show OTR expression already prenatally in both mice and rats (Figure 1). Moreover, OTR expression in this limbic region is found in every developmental period, which makes the amygdala a probable important, chronic effector for the functions of OT in the brain.

### 2.3. The Functions of Oxytocin in the Brain

The wide-spread nature of the central OT system relates to with the diversity of functions identified for OT in the brain [28,30]. These include the traditional links with lactation, parturition and social behaviors such as social bonding and maternal behavior and aggression, but also extend to non-social behaviors such as anxiety, fear, decision making and memory [29,30,80,81,82,83]. Most of these functions are directed by central OT release, but some effects of OT, for example pain control, arrive from a combination of central OT projections onto the spinal cord, and hormonal OT circulating in the bloodstream [84]. It has further been shown that the OT system can self-adapt by increasing or decreasing the expression of OTR or density of OT fibers in response to environmental stimuli (for example, mother-infant bonding increases OTR expression in areas involved in social behavior [54]). However, while the density of OT connections can be variable, the spatial profile of connections appears to remain unchanged upon stimulation [30].

There are specific functional ‘subsystems’ within the OT network [85]. Recently, input-output wiring diagrams of OT neurons revealed 9 distinct circuit-specific functions for OT in the adult mouse brain [86]. These include internal state control, including attention and threat, somatic visceral control including pain and sensory regulation, and cognitive control including learning and value assessment [85,86]. We refer to [86] for an overview of the specific brain regions involved in each of the 9 functional circuits of OT. Hypothalamic OT neurons show axonal projections to most of the forebrain regions in the adult rat [30]. This includes the limbic system, olfactory system, basal ganglia and cortical areas. OT projections were also found in the hippocampus, which is functionally linked to the discrimination of social stimuli [87] and long-term social memory [88,89]. Social behavior and social decision-making are one of the strongest functional attributes known to the OT system [28,90]. Various OTR-enriched regions have been linked to aggression, sexual behavior, maternal care and pain inhibition (medial amygdala, lateral septum, stria terminalis, hypothalamus and periaqueductal and central gray) and the processing of salient social stimuli (basolateral amygdala, stria terminalis, lateral septum, nucleus accumbens, striatum, ventral pallidum, ventral tegmental area and prefrontal cortex) (reviewed in [60]).

In addition to the contributions of OT to functional brain development, OT also plays a role in structural brain development. OT contributes to neurite growth and the formation of neural circuits [91]. Moreover, experimental research in OT-knockout mice has shown that OT is required for the promotion of excitatory synaptic transmission during sensory cortical development [47]. Additionally, OTR expression in hippocampal CA3 has been associated with the promotion of cell survival and development of newly formed dentate granule cells [92]. Another major contribution of OT to normal brain development concerns the GABA switch. The GABA switch refers to the change from excitation to inhibition of the GABA neurotransmitter during the first week of life, which is important for the maturation of neuronal network functioning [93]. The large amount of OT that is released during parturition helps facilitate this switch [94,95]. Neonatal complications that reduce OT release, such as neonatal maternal separation, have shown to delay the GABA-switch, which induces an imbalance to excitation/inhibition, that is characteristic of many neurodevelopmental disorders [96,97].

#### Major Functional Part of the Oxytocin Network: The Amygdala

One area that has been extensively linked to OT functioning is the amygdala. Indeed, the study by Knobloch and colleagues [30] found a high number of OT fibers in the central and medial regions of the adult rat amygdala. This has been shown in humans as well [56], including on the transcriptome level, where amygdala-associated cognitive terms “anxiety”, “fear” and “emotional” showed a 97.5% stronger correlation with OTR mRNA expression than any other genes in a 20.737 gene dataset [90]. The amygdala plays a major role in social decision making [98,99]. Amygdala damage impairs social behavior, and OT injection to the basolateral amygdala stimulates prosocial behavior in primates [98]. Moreover, intranasal OT administration in humans increases social cognitive processing and emotional empathy, which is amygdala-dependent [100,101]. On a network level, the increased sociability following OT administration was associated with decreased amygdala reactivity to negative stimuli, but increased coupling between the right amygdala with the posterior cingulate cortex and insula for positive stimuli [100].

In line with this function of social decision making, there has been much support underlining the role of OT in modulating anxiety and fear via its effect on the central amygdala [30,31,80]. Optogenetic activation of OT neurons has been shown to decrease freezing responses in rats that were previously fear-conditioned, which was associated with enhanced glutamate co-release by the projecting OT terminals in the amygdala [29,30]. The OT-induced anxiolytic effect is further emphasized by studies that showed a reduction in fear memory retrieval after OT was infused into the central amygdala before the fear acquisition phase [31]. Regional specificity within the central amygdala has further identified “Fear-OFF” neurons in the lateral division, which are protein kinase C-δ-expressing GABAergic neurons that decrease firing upon fearful events [102]. Notably, OTR expression has been found in more than half of this neuronal population [102,103]. Moreover, it appears that astrocytes play a role in the OT-induced anxiolytic effect in the central amygdala. Gain- and loss-of-function paradigms demonstrated that a particular local astrocyte subtype positively reinforces the effects of OT on the central amygdala, which challenges the long-held idea that OT acts exclusively on neurons to modulate emotional states [32].

These data show that the contribution of OT to brain development is both functional and structural, because it helps form functional pathways between regions involved in important behaviors, but it also contributes to the growth and maturation of individual neurons. Given this importance of OT for brain development, it is likely that OT functioning is affected when perinatal complications cause the brain to develop abnormally.

## 3. Oxytocin Functioning in Abnormal Brain Development

The developing brain can be challenged by perinatal injury, trauma or bacterial and viral infection. As a consequence, physiological processes related to OT functioning can be impaired and delayed, which can have a severe impact on later cognitive and behavioral functioning.

### 3.1. Neuroinflammation as a Common Factor of Perinatal Complications

The major types of perinatal complications include preterm birth, intra-uterine growth restriction, hypoxia, neonatal stroke, and the potential resulting neurodevelopmental disorders including ASD and attention-deficit-hyperactivity disorder [104,105,106]. What these complications have in common is that they are accompanied by a specific change in the brain environment, namely the brain changes into a neuroinflammatory state [7,107]. Neuroinflammation can be induced in reaction to environmental toxins, bacteria, or neuronal injury or death, and it makes the brain change into a state of attack [108]. The cells responsible for initiating the neuroinflammatory response are microglia, the brain’s resident immune cells that make up 10% of the glial cell population in the cerebral cortex [109]. Microglia are dynamic glial cells that can change their phenotype and function in response to the neural environment [110]. Microglial states have been traditionally categorized into the M1 and M2 types, but recent recommendations reject this over-simplifying classification (we refer to [111] for an overview and guidelines on the use of microglial nomenclature). In their basal homeostatic state, microglia survey the brain and contribute to maintaining optimal conditions for neuronal functioning. When sensing an intrusive factor or neuronal injury, microglia become reactive and change their shape [112,113]. They start secreting pro-inflammatory cytokines that further activate the inflammatory response, which is directed towards eliminating the intruder, restricting the jury, and preserving the healthy tissue [114]. The second major cell type that is involved in the neuroinflammatory response are astrocytes. Astrocytes are star-shaped cells that make up 20–40% of the cerebral glial cell population [109]. They can be activated by environmental toxins and brain injury, and also by reactive microglia [115]. In their reactive form, astrocytes can induce neuronal death and further amplify the inflammatory response by activating other astrocytes but also distant microglial cells [115,116]. Even though the inflammatory response is effective at its initial goal of isolating and reducing the toxic effects on the brain, the activation of microglia often leads to an over-activation, in which the inflammatory response exceeds its beneficial effects and becomes damaging for the healthy neighboring tissue [108,117]. Needless to say, when this occurs in such an early phase as the perinatal period, the impact on brain development is immense. Indeed, infants who suffer from perinatal complications show an increase in inflammatory cytokines (a marker of neuroinflammation) [118,119]. This effect is especially apparent in the amygdala [120]. Thus, the neuroinflammatory response is a major contributing link between perinatal brain injury and the long-term cognitive and behavioral problems that can arise.

### 3.2. Altered Oxytocin Functioning in Neuroinflammation and Other Neurodevelopmental Disorders

Following the establishment that neuroinflammation plays a central role in the pathogenesis of perinatal brain damage, the question arises of how OT acts in this sequence. OT is an important player in brain development at both the functional and structural level, and neuroinflammation has been shown to interfere with this system. A meta-analysis of 18 studies comprising 1422 participants found lower levels of endogenous OT in the blood plasma or saliva of children with ASD compared to neurotypical controls [121]. This effect was especially prominent in young children (<9 years of age), and was not found in adolescents and adults with ASD. This suggests that the modulation of OT in neurodevelopmental disorders (NDDs) has a developmental time-course, yet it should be noted that no clear conclusions about central OT can be drawn from peripheral OT data. Nevertheless, it raises the notion that early life disruptions to the OT system have a larger effect on brain development than later-life changes, which is found as well in animal models of ASD, both genetic and environmentally induced. Decreased CSF-OT and cerebral-OT concentrations and a lower number of OT neurons have been found in neonatal mice that are knock-out for the ASD-associated gene Magel2 [67,122], as well as in adolescent rats that were exposed to valproic acid [123]. This decrease in neonatal OT was attributed to the mature OT form, with no changes to OT-prohormone levels [67]. Interestingly, levels of intermediate OT were elevated in the PVN of Magel2-KO mice [122], which suggests a deficiency or delay in the transformation of intermediate OT to mature OT in ASD animal models. It appears, moreover, that the decrease in postnatal OT found in models of ASD is followed by increased OT at adult age [122,124]. This effect was particularly apparent in the central and medial amygdala, which showed, respectively a 200% and 300% increase in OT concentrations in Magel2-KO mice compared to WT animals.

In addition to these findings, studies on neuroinflammation found that interleukin-1-β (IL-1β) or lipopolysaccharide (LPS) administration to adult male rats was accompanied by increased plasma OT levels, OT release from the SON, and increased activity of immediate-early gene c-FOS in all regions where OT is synthesized [40]. Similar increases in OT activity were found in MG6 microglia cells following LPS stimulation [125]. Finally, in a rat model of fetal growth restriction, increased pro-inflammatory cytokine levels and microglial activation were accompanied by substantially downregulated hypothalamic OT levels in the neonatal pups [126].

Taken together, these findings support a developmental, anatomical and functional deregulation of the OT system in inflammation-related NDDs, with a decreased concentration of mature OT in the postnatal period, followed by increased OT levels in adulthood. This led some researchers to hypothesize that the social deficits seen in ASD models such as Magel2-KO is due to deficiency in the hippocampal OT system [122,124]. The amygdala is a location that seems especially affected by neuroinflammation, which is accompanied by alterations to OT functioning.

## 4. Experimental Studies on Neonatal Oxytocin Treatment

Given the change in OT levels and OT system functioning seen in neuroinflammatory-related states, the question becomes if neonatal OT could be a therapeutic option for neuroinflammation and, subsequently, if this could have a protective effect on the developing brain. The majority of data come from experimental studies using animal models of neuroinflammation and its related developmental disorders. These models simulate part or most of the inflammation symptoms seen in human infants, including the detrimental effects on brain structure and function, as well as more long-term behavioral deficits. There are generally three types of models used to study neuroinflammation in the developing brain. The first category includes models with direct exposure to pro-inflammatory factors such as IL-1β or LPS during the neonatal period, either systemically into the blood stream or locally into the brain [127,128]. The second category comprises indirect models of neuroinflammation which simulate the perinatal complications that are known to go hand in hand with neuroinflammatory responses. These include models of intra-uterine growth restriction (IUGR), hypoxic-ischemic insult (HI), pediatric traumatic brain injury (TBI) or neonatal stroke, sometimes combined with injections of pro-inflammatory cytokines [107,126,129]. The third category of experimental neuroinflammation models includes models that focus on the neurodevelopmental disorders that can arise from perinatal complications, such as ASD.

### Neonatal Oxytocin Treatment to Alleviate Inflammation-Induced Brain Injury

Multiple studies have tested neonatal OT as a therapy in neuroinflammation models and neuroinflammation-related models (Table 1). Administration of OTR-agonist carbetocin at P1 and P2 showed to rescue the inflammatory phenotype in rat pups that were exposed to a double-hit model of IUGR [126]. Specifically, carbetocin reversed the upregulation of pro-inflammatory factors IL-6, IL-1β, TNF-α, and iNOS in the rat cortex at P4, and it rescued the decreases in white matter development at P10. The protective effects of OT translated to a behavioral level, as carbetocin-treated IUGR pups showed no increase in anxious behavior or incapacity for spatial memory, compared to their saline-treated IUGR littermates [126]. Similar results were found in a rat model of neonatal hypercapnic-hypoxia. This model caused an increase in TNF-α and astrogliosis in the CA1, compared to healthy pups [130]. Neonatal OT injections rescued these effects. Although they did not assess the neonatal age, additional insights into the effects of OT on neuroinflammation can be given by Yuan and colleagues [131]. In adult BALB/C males, pre-treatment with a single-hit of intranasal OT before LPS injection markedly reduced microglial activation and pro-inflammatory factor levels. More specifically, LPS increased TNF-α, IL-1β, COX-2 and iNOS protein and mRNA production, TNF- α/Iba1 immunoreactivity, and the number of microglia with activated cell-morphology. All these increases were significantly reversed by OT administration, which was shown both in vitro and in vivo [131].

Given the aforementioned changes in the OT system seen in ASD, multiple studies have tried neonatal OT as a treatment for the ASD-like phenotype. Valproate (VPA)-induced ASD models in mice showed an increase in pro-inflammatory cytokine expression and increased oxidative stress response in regions including the amygdala [132]. Intranasal OT treatment at the juvenile age reversed this inflammatory phenotype and rescued the autism-like behavior [132]. Similar beneficial effects on ASD-like behavior were found following chronic OT treatment between P0 and P6 in VPA-exposed rats, both in the short-term (P7) and at juvenile age (P35-P40) [123]. Using the Magel2-KO genetic mouse model of ASD, several studies have shown that peripheral OT injections during the neonatal period restore the deficits in social, cognitive and socio-cognitive behavior at 4 months of age [122,124]. It should be noted that genetic ASD models often do not use a pro-inflammatory instigator for its disease phenotype, which makes it difficult to see if the effect of OT on behavioral outcomes is associated with changes in inflammatory states. There is, however, much support for the implication of neuroinflammation and microglial cell activation in ASD. One genetic ASD paradigm showed a reduction of microglial-Iba1 immunoreactivity in the first postnatal week in the basolateral amygdala region, which is one of the main regions implicated in the executing behaviors impaired in ASD [133]. Moreover, increased plasma cytokine levels of interferon-γ, IL-1β, IL-6, and TNF-α have been strongly linked to ASD diagnosis in school-aged children [134,135,136]. Clinical trials showed a better effect size of risperidone, a classical ASD drug, when it was co-administered with other drugs targeting cytokine inhibition [137,138,139,140]. Therefore, although not all OT-ASD studies assess the direct link with neuroinflammation, the body of available work does support a strong link on the one hand between OT and developmental outcome in ASD, and on the other hand between neuroinflammation and ASD.

## 5. The Underlying Mechanisms for the Neuroprotective Effect of Oxytocin in Neonatal Neuroinflammation: Astrocytes and Microglia

Due to the intricate interplay between microglia and astrocytes in neuroinflammation, there are several possibilities of how OT reduces the inflammatory response. A major question is whether OT affects mainly microglia, i.e., the ‘organizers’ of the inflammatory response, or more the ‘executors’ of the inflammatory-induced neurotoxic effects, which are the astrocytes. In the next part, we will discuss potential mechanisms through which OT can reduce neonatal inflammatory response.

### 5.1. Oxytocin and Astrocytes

It is plausible that OT dampens the inflammatory response through astrocytes because the presence of OTR expression on astrocyte soma and processes has been demonstrated in numerous brain regions, including in the hypothalamus, hippocampus, frontal cortex, auditory cortex, ventral striatum and the amygdala [141,142] (reviewed in [143]). Moreover, a specific OTR-positive subpopulation of astrocytes has been identified in the central amygdala, characterized by a larger cell volume and surface area and increased process length [32]. Electrophysiological experiments revealed depolarization of the astrocyte membrane potential and an increase in calcium release upon activation of the OTR [144,145]. Furthermore, selective OTR knockout in GFAP-positive cells showed a decreased response in astrocyte calcium signaling to OT application, which supports that the effect of OT on astrocytes is indeed attributed to astrocytic OTR [32]. OTR activation in astrocytes has the possibility to increase local astrocyte network activity, but it can also change neuronal excitability and amygdala neuronal network output [32].

In the neuroinflammatory response, astrocytes are activated by microglia via secretion of pro-inflammatory cytokines IL-1α, TNF-α and C1q [146]. In their reactive state, astrocytes facilitate the neurotoxicity and apoptosis of mature neurons and oligodendrocytes via the secretion of soluble neurotoxins [146]. Moreover, they lose their normal phagocytic activity, which is detrimental to the optimization of synaptic transmission. Reactive astrocytes also amplify the inflammatory response [115]. Through astrocytic calcium waves, they can activate not only other astrocytes but also microglial cells in distant locations, thereby facilitating a neurotoxic response in brain tissue that does not show any damage [115]. Notably, the reactive astrocyte phenotype is associated with a decrease in the expression of OTR [142,147]. After the initial inflammatory phase, astrocytes start shifting from a pro-inflammatory to an anti-inflammatory phenotype. The latter is more focused on remodeling of the damaged tissue and the neurovascular unit, which it achieves via secretion of beneficial factors such as CLCF1 (cardiotrophin-like cytokine factor 1), LIF (hypoxia induce factor), IL-6, IL-10 and thrombospondins [116].

This distinction between the pro- and anti-inflammatory phenotypes is an example of the highly heterogeneous nature of the astrocyte population. Using single-cell RNA sequencing, Hasel and colleagues [147] identified 10 such separate astrocyte subgroups, with cell subpopulations showing different responses to inflammatory stimulation. Two subgroups showed a particularly strong reaction to LPS stimulation. One of these, “cluster 8” astrocytes, showed a strong, early response to the LPS stimulation which gradually degraded at 24 h and 72 h post-insult. The other cluster, “cluster 4” astrocytes, showed a peak of activation at 24 h post-stimulation [147]. The fact that astrocytes show a heterogeneous response to an inflammatory stimulus suggests that the effect of OT on astrocytes during inflammation could also be heterogeneous, and could differ per astrocyte subpopulation. Consequently, there are likely two main ways through which OT can reduce the adverse neuroinflammatory-effects via astrocytes: (1) by deactivating the pro-inflammatory astrocyte phenotype, (either the early responding cluster, late responding cluster, or both), or (2) by accelerating the shift to the neuroprotective anti-inflammatory astrocyte phenotype.

#### 5.1.1. Shift to Neuroprotective Astrocytes

In vitro work has shown that after activation by IL-1α, TNF-α and C1q, reactive primary astrocytes do not passively revert back to a resting-state type once the pro-inflammatory cytokines are removed from the culture medium [146]. Rather, they require active stimulation by anti-inflammatory cytokines such as transforming growth factor-β (TGF-β) or fibroblast growth factor (FGF) to decrease their reactivity [146]. Interestingly, TGF-β can be produced by astrocytes, but it is also secreted by microglia during inflammation [116]. In microglia, TGF-β is likely involved in the microglial-organized shift from the toxic astrocyte phenotype to the remodeling astrocyte subtype [116]. A link between TGF-β and OT has been found in astrocytes purified from E16 rat embryos and cultured for 12 days [148]. Namely, TGF-β increased the binding of astrocyte OTR and astrocyte OTR mRNA levels [148]. This suggests that increased astrocytic TGF-β signaling could be a target for potential OT treatment. Unfortunately, this association has not been studied yet to detail in an inflamed neural environment. Data from cultured explants of fetal membranes, however, showed that increased OT levels were associated with increased TGF-β1 levels [149], which suggests that the association between OT and TGF-β could also persist in other inflammatory environments, such as the brain.

In the healthy adult male rat, OT administration showed to decrease astrocyte-specific GFAP immunoreactivity in the SON [145]. Using specific protein inhibitors, this effect was associated with the inhibition of ERK1/2 kinase and activation of protein kinase A (PKA), which suggests that the downstream effects of OT in astrocytes involve the MAPK pathway [143]. However, it is not clear if this pathway of effect persists in neuroinflammatory states. That said, it was found in a mouse model of chronic migraine that intranasal OT treatment reduced the activity of the PKA/CREB pathway [150], which forms some support that the OT-MAPK-pathway of effect extends to physiological states.

#### 5.1.2. Astrocyte Process Retraction to Reduce Neurotoxic Effects

Another option for OT to execute its anti-inflammatory effect is to inhibit the neurotoxic events mediated by reactive astrocytes. Due to their presence at the tripartite synapse, astrocytes are in close contact with neurons (1 astrocyte can even link with up to 100.000 synapses [116]). This proximity stimulates the apoptotic effects of reactive astrocytes on healthy neurons via the release of neurotoxins [146]. It has been shown in healthy conditions in the hypothalamus that OT application to astrocytes triggers the retraction of their processes from the synapse [143]. This OT-mediated change in astrocyte cytoskeleton is likely brought about by an effect on microtubule/actin dynamics [143,151], and it is linked with beneficial effects on anxiety in rats [32]. Similar effects were found in astrocyte cultures purified from Wistar rats, where the effects of OT on astrocytic gap-junction coupling and cytoskeletal remodeling were conveyed via PKC and MEK1/2 signaling [152]. Moreover, it was found that Gem, and the Gem-Sp1 pathway are required and sufficient for the OT-mediated effects on astrocytes [152]. These reports raise the possibility that in neuroinflammatory states, OT-mediated retraction of astrocyte processes from the synapse/gap-junction could form a spatial hurdle for astrocytes to execute their neurotoxic effect, for it is plausible that a spatial increase to the synapse consequentially decreases the amount of toxic gliotransmitter that reaches the surrounding neurons [143].

### 5.2. Oxytocin and Microglia

Having a high sensitivity to the neural environment, microglia are the general ‘organizational force’ behind the inflammatory response. Many studies have shown that OT treatment induces a shift in microglial phenotype and reactivity during the inflammatory response [126,131,132,153,154]. In vitro studies show that OT pre-treatment reduces the LPS-induced increased production of pro-inflammatory cytokines from primary microglia and macrophages [131,155]. Moreover, OTR knock-out mice show increased microglial activation in the medial amygdala compared to WT mice, as evidenced by an increased Iba1-immunoreactivity, and a stronger presence of amoeboid-shaped cells [156]. There are several possible mechanisms underlying the effect of OT on microglial reactivity during neuroinflammation. However, up to date no direct effect of OT treatment on early life microglia has been demonstrated in vivo.

#### 5.2.1. Reducing Phagocytic Activity via NADPH/ROS Signaling

Being phagocytic cells, microglia express high levels of superoxide-producing NADPH oxidases, a family of enzymes involved in the production of reactive oxygen species (ROS). The neurotoxicity mediated by activated microglia is highly dependent on NADPH oxidase [108]. It has been shown that treatment of macrophages and THP-1 monocytes with OT markedly reduced the production of NADPH oxidase-dependent superoxide [154]. Additionally, this showed to reduce the production of the pro-inflammatory cytokine IL-6. Despite it not being shown directly in microglia, these data support that a similar mechanism of effect could exist for OT on microglial cells.

#### 5.2.2. Regulating Cytokine Secretion via the NF-кB/MAPK Pathway

An important factor in the production of microglial pro-inflammatory cytokines are mitogen-activated protein kinases (MAPKs), including ERK1/2, p38 and JNK [157]. The anti-inflammatory effect of OT seen on microglia is thus possibly via the MAPK pathway. Indeed, the OT-induced decrease in pro-inflammatory cytokine expression in cultures of a BV-2 murine microglial cell line and primary microglia was associated with decreased phosphorylated ERK1/2 and p38 MAPK protein levels [131]. However, in an LPS-stimulated MG6 microglial cell line, OT treatment did not affect p38 MAPK phosphorylation levels [153], which suggests there are methodological differences between cell lines and primary microglia. Closely related to MAPK, the NF-кB protein is a transcription factor that regulates pro-inflammatory cytokine production in microglia. It has been shown that NF-кB inhibition following neonatal hypoxic-ischemia in rats greatly reduced brain damage [158]. LPS-stimulation of primary microglia [131], and the MG6 microglia cell line [153] increased the phosphorylation of the NF-кB protein, but OT pre-treatment had no effect in either study. However, a neonatal maternal separation model in rats showed that, compared to vehicle, OT pre-treatment reversed the increase in GFAP signal and TLR4/NF-кB signaling [159]. Moreover, in a mouse model of early stage Alzheimer’s disease, which is a disease model associated with increased neuroinflammation levels, it was found that OT release inhibited microglial activation and cytokine secretion by blocking the ERK/p38 MAPK and COX2/iNOS NF-кB signaling pathways [160]. These data suggest that the NF-кB/MAPK pathway could be an important factor in the neuroprotective effect of OT on microglia, but they also underline the discrepancy between in vitro and in vivo studies. In vivo experiments would have the upper hand, but data at this experimental level are lacking for neuroinflammation-specific physiological environments.

#### 5.2.3. ER Stress-Related eIF-2a-ATF4 Pathway

Proteins are folded and undergo post-translational modifications in the endoplasmic reticulum (ER). If cells experience ER stress, a coping-mechanism is activated to provide homeostatic balance [161]. This mechanism is activated through ER stress sensors, including ATF4, IRE1 and PERK. ER stress is known to increase the inflammatory response, and stimulate the production of pro-inflammatory cytokines from macrophages [153,161,162]. The role of ER on inflammation in microglia is further supported by reports that LPS-stimulation in MG6 microglia cells increases phosphorylation levels of elevated eukaryotic initiation factor-2a (eIF-2a), which is a factor targeted by the ER-stress sensor PERK [153]. OT treatment showed to suppress this increased phosphorylation, an effect that was blocked by OTR antagonist administration [153]. Moreover, ATF4 is a transcription factor that is translated by eIF-2a. ATF4 levels were increased and subsequently rescued by LPS and OT pre-treatment, respectively [153]. Downstream effects of ATF4 activation by LPS stimulation include an increase in transcriptional targets CHOP, GADD34 and caspase-1/11, which is involved in the production of IL-1β. OT has been shown to suppress the increase in these factors [153]. Together, this work suggests that OT dampens the inflammatory response by acting through the eIF-2a-ATF4 ER stress-related pathway in microglia, which reduces the production of pro-inflammatory cytokines [153].

### 5.3. Glial Crosstalk in the Effect of Oxytocin: The Astrocyte Versus Microglia Battle

Given the potential effects of OT on both microglia and astrocyte functioning, and the fact that microglia and astrocytes can bidirectionally influence each other’s activity level through cytokine/chemokine signaling, an interesting question arises which glial cell type contributes more to the anti-inflammatory effects of OT. LPS injections to adult mice showed to increase TNF-α reactivity levels of both astrocytes and microglia, but pre-treatment with OT only reversed the TNF-α immunoreactivity of microglia and did not affect astrocytes [131]. This suggests that, at least in adults, the anti-inflammatory effect of OT may be independent of astrocytic reactivity [131]. However, opposite results have been found, as the increased proliferation of CA1 GFAP-positive cells seen in a mouse model of neonatal hypercapnia-hypoxia injury was reversed by pediatric OT treatment [130]. A potential explanation could thus be that OT treatment during inflammatory states affects the number of astrocytes, but not their reactive phenotype. Furthermore, OT injections between P2 and P6 in healthy rats showed to increase in the expression of hippocampal GFAP mRNA and protein 2 months later but showed no change in CD68 levels, which is a marker of microglial phagocytic activity [163]. This could suggest that the respective effects of OT on astrocytes and microglia are dependent on the presence or absence of inflammation [161]. Consistently, the severity of inflammation likely determines the ability of astrocytes to inhibit microglia, as astrocytes have been shown unable to inhibit the microglial production of pro-inflammatory nitric oxide (NO) when inflammation levels are too strong [115]. As such, an OT-induced decrease in inflammation level, regardless of mediated through which glial cell type, may allow astrocytes to better inhibit microglia, but further assessment is required to test these speculations. Another factor in unraveling the effect of OT on the astroglial interaction during inflammation could be the timing of treatment. Liddelow and colleagues [146] showed that after the axotomy of retinal ganglion cells, inhibition of reactive astrocytes improved cell death, but inhibition of microglia had no effect. This raises the possibility that once a complication/injury has taken place, the OT-mediated effect on astrocytes is more effective than microglia in protecting the developing brain from inflammation. However, a lack of direct evidence exists, and these premises require more direct investigation.

#### 5.3.1. Oxytocin Acts to Beneficially Influence the Control of Astrocytes on Microglia

Nevertheless, there are some data available about specific ways that astrocytes could relay the anti-inflammatory effect of OT on to microglia cell activity. This includes both an increase in the astrocytic ‘silencing’ control on reactive microglia, and a decrease in the stimulation of microglia reactivity by astrocytes. As mentioned earlier, one of the anti-inflammatory cytokines produced by astrocytes is TGF-β, and there is support suggesting a positive link between TGF-β and OT [148]. Yet, in addition to influencing astrocytes, TGF-β can also act on microglial cells by binding to the TGF-β receptor type II, which phosphorylates TGF-β receptor type I [164]. TGF-β secretion has been shown to reduce microglial activation, evidenced by a reduction in microglial production of pro-inflammatory cytokines, NO and oxygen-free radicals [115]. This was associated with increased activity of NF-кb and IL-1 in microglia [115]. In vitro work has further shown that in baseline conditions, astrocytes in collaboration with neurons use TGF-β2 to form an immune-regulatory control on microglia, which represses the microglial response to weak inflammatory stimuli [165]. Loss of TGF-β signaling in microglia via KO of the TGFB2-receptor increased microglial activation on a morphological and transcriptomic level [164]. As such, the positive associations between OT and TGF-β can, in addition to directly reducing astrocyte reactivity, also have a dampening effect on microglial reactivity through astrocyte signaling.

The second means of indirect effect could involve an OT-induced reduction in the stimulating effect of astrocytes on microglia phagocytosis. In response to local injury, astrocytes release ATP, which activates local microglia and initiates microglial proliferation [166]. Moreover, astrocytic ATP facilitates microglial phagocytosis of neurons. This can be blocked by apyrase, an ATP-degrading enzyme, and inhibition of the P2X7R on microglia [166]. Inhibition of microglial P2X7R reduced IL-1β and TNF-α protein and mRNA expression in rats exposed to acute myocardial infarction [167]. It also affected the function of OT, as P2X7R inhibition decreased the number of OT cells in the PVN. An additional factor involved in the regulation of microglial phagocytosis is C1q. Due to their close proximity to neurons, astrocytes can influence pre-synaptic neurons to produce C1q, which interacts with the microglial C3a receptor to induce phagocytosis [165]. Another signaling molecule through which astrocytes can activate microglia in inflammation is astrocyte-derived chemokine (C-C motif) ligand 7 (CCL7). It was shown that knockout of CCL7 decreased neuroinflammation in a rat model of TBI, which led to improved brain morphology and neurological functioning [168]. On a molecular level, CCL7 deletion decreased Iba-1 and GFAP immunoreactivity, as well as TNF-α, IL-1β, IL-6 and IL-18 production in the cortex and serum [168]. Thus, there are several mechanisms through which astrocytes can influence microglial phagocytosis during neuroinflammation. These mechanisms pose interesting targets for the protective effects of OT, but more research is required to test these hypotheses.

#### 5.3.2. How Does Oxytocin Reach Microglia: The Microglial Oxytocin Receptor Debate

Despite the evidence that OT affects microglial reactivity in neuroinflammation, it is far less clear how OT actually reaches microglia and changes its phenotypes. Specifically, whether this effect is indirect, as mediated through astrocytes (described above), or if it is a direct effect in the form of microglial OTR expression (Figure 2). Unlike astrocytes, the presence of OTR expression on microglia has not received the same body of support. Some studies do find microglial OTR expression but not already in the neonatal age, not on all experimental levels (In vitro vs. in vivo) or all conditions (basal vs. inflammatory-stimulated) (Table 2).

The studies that do report OTR expression on microglia are predominantly in vitro studies [131,142,153,155]. Yet, LPS stimulation of in vitro microglia seems to have a highly variable effect on microglial OTR expression. Namely, LPS stimulation of P5 primary microglia induced a decrease in OTR mRNA levels as assessed with RNA Sequencing [142]. Yet, LPS stimulation of P1-P2 primary microglia caused an increase in OTR mRNA and protein levels [131]. Macrophages purified from 3-month-old C57BL/6 mice, however, showed no change in OTR expression in inflammatory states [155]. This raises the question of whether the developmental stage of the subject/microglial cell (line) in vitro influences how OT reacts to neuroinflammatory states. Moreover, the discrepancies between these studies could be explained by the origin of collected microglia (whole brain for Guttenplan and colleagues [142], or cerebral cortex for Yuan and colleagues [131]) and intensity of LPS stimulation (50 ng/mL for Guttenplan and colleagues [142], and a ten-time increase in Yuan and colleagues [131], with 500 ng/mL). More work is required to map the influencing factors between inflammation and microglial OTR expression in vitro.

To our knowledge, little to no OTR expression on microglia has yet been revealed in vivo. Available databases of in vivo microglia single-cell RNA Seq analysis (see http://www.brainrnaseq.org/, http://www.microgliasinglecell.com/, https://myeloidsc.appspot.com/, all accessed on 1 October 2022) show a trend where OTR transcripts are little-to not detected in the embryo and early postnatal ages, but only in adult mice (Table 2) [169,170,171,172,173]. Moreover, if OTR mRNA *is* found, it is with very low read counts, below the cutoff threshold of quality assessment measures [169,171,173]. Karelina and colleagues [174] did find OTR expression in microglia in vivo. Using flow cytometry, they found 16% of microglia expressing OTR. However, they used only adult mice. In contrast, a study that harvested microglia from E16 rat embryo’s found no OTR binding after 12 days in culture [148]. This suggests that microglial OTR expression cannot necessarily be warranted in neonatal animals, and should be investigated in more detail. Consequently, it could be that the anti-inflammatory effects of neonatal OT treatment seen in experimental in vivo studies are, at least in the early phase of life, more due to the effect of OT on astrocytes, or that the eventual effect of OT on microglia in the neuroinflammatory response is an indirect effect through astrocytes.

**Table 2 cells-11-03899-t002:** **Overview of studies assessing oxytocin receptor expression in microglia in healthy and inflammatory states.** LPS = Lipopolysaccharide; TNF-α = Tumor necrosis factor alpha; IL-1β = Interleukin-1 beta; COX2 = cyclooxygenase-2; iNOS = Nitric oxide synthase isoform; RT-PCR = real-time polymerase chain reaction; SD = standard deviation; FPKM = fragments per kilobase of exon per million mapped fragments; CPM = counts per million reads mapped; kDa = kilodalton; FACS = Fluorescence-activated Cell Sorting.

Study; Journal	Species; Strain; Sex	Age of Assessment	Assessment Method	In Vivo or In Vitro?	OTR Expression Found?	Type of Inflammation	Effect of OT Treatment
[142] Nat Commun.	Primary microglia from C57BL/6 mice; sex unspecified	P5	Bulk RNA Seq: Illumina HiSeq 4000	In vitro; cultured for 7 days	yes (Mean = 1.139 SD = 0.904 FPKM)	LPS stimulation (50 ng/mL) for 24 h	LPS decreased OTR expression in C57Bl/6 mice (Mean = 0.486 SD = 0.486 FPKM)
[155] Am J Physiol Endocrinol Metab	(1) Primary macrophages from male and female humans (2) Peritoneal macrophages from male C57BL/6 mice	Human: Not disclosed Mouse: 12 weeks	qPCR and Western blot	In vitro; cultured for 7 days	Human: yes Mouse: yes	LPS stimulation (100 ng/mL) for 6 h	Human: LPS increased OTR protein expression of both 67 kDa and 46 kDa forms. Mouse: no effect
[153] Cells	Microglial cell line MG6 from C57BL/6 mice; sex unspecified	Not applicable (cell line)	RT-qPCR; Western blot; ELISA	In vitro; cultured for 2–3 days; including 30 min OT pretreatment before inflammation	Not assessed directly, but OTR antagonist L-371,257 pre-treatment reversed the anti-inflammatory effects of OT	LPS stimulation (100 ng/mL) for 24 h	OT treatment suppressed proinflammatory cytokine production in LPS-stimulated MG6 microglia
[131] J Neuro-inflammation	BV-2 cells and primary microglia; sex unspecified	P1-P2	RT-qPCR; immuno-histochemistry	In vitro; cultured for 21 days (14 days mixed-glia culture, 7 days microglia culture); including 2 h OT pretreatment before inflammation	BV-2 cells: yes Primary microglia: yes	LPS stimulation (500 ng/mL) for 24 h	LPS increased OTR mRNA expression in BV-2 cells and primary microglia. OT pre-treatment suppressed LPS-stimulated expression of TNF-α, IL-1β, COX-2 and iNOS at the protein and transcriptome level.
[174] Stroke	CD11b-positive microglia from male C57BL/6 mice	Adult	RT-PCR and flow cytometry	In vivo and in vitro In vitro: primary microglia, 2 h pre-treatment with OT or OT antagonist	In vivo: 16% of CD11b-positive cells expressed OTR In vitro: Not assessed directly, but OTR antagonist application blocked the anti-inflammatory effect of OT	LPS stimulation (1 μg/mL) for 22 h	In vivo: not assessed. In vitro: OT incubation before LPS stimulation attenuated major histocompatibility complex class II expression.
[169] Proc Natl Acad Sci U S A	FACS-sorted microglia from C57BL/6 mice, male and female	E17, P7, P14, P21 and P60	Bulk RNA Seq. Database: http://www.brainrnaseq.org/, accessed on 1 October 2022	In vivo	E17: no P7: no P14: no P21: no P60: yes (Mean = 0.120 SD = 0.120 FPKM)	LPS stimulation (5 mg/kg, single intraperitoneal injection at P60)	LPS treatment depleted OTR expression at P60 (Mean = 0, SD = 0 FPKM)
[173] J Neurosc	Microglia and macrophages from Tie2–EGFP transgenic mice; sex unspecified	P7	Bulk RNA Seq: Illumina HiSeq 2000. Database: http://www.brainrnaseq.org/, accessed on 1 October 2022	In vivo	Yes: (Mean = 0.1 SD = 0 FPKM)	None	Not applicable
[170] Immunity	FACS-sorted microglia from C57BL/6 mice, male and female	E14.5, P4/P5, P30, P100, P450	Single cell RNA Seq: Illumina NextSeq500 Database: http://www.microgliasinglecell.com/, accessed on 1 October 2022	In vivo	Not detected at any time-point.	None	Not applicable
[171] Neuron	FACS-sorted microglia from male C57BL/6 mice	E14.5 (whole brain), P7 and P60	Single cell RNA Seq: Smart-seq2. Database: https://myeloidsc.appspot.com/, accessed on 1 October 2022	In vivo	P60 clusters: 0–0.1 CPM P7 clusters: 0–0.1 CPM Embryo cluster: 0–0.1 CPM (Did not reach cut-off threshold of >2CPM)	None	Not applicable
[172] Nat. Neurosci.	FACS-sorted microglia from male C57BL/6 mice	P90	Single cell RNA Seq: Illumina NextSeq500	In vivo	No	None	Not applicable

Another option is that the effect of OT on microglia during neuroinflammation goes through the modulation of additional neurotransmitter systems, such as serotonin (Figure 2). It has been shown in mice that serotonin can modulate microglial reactivity via the microglial 5-HT2b receptor, and that conditional knock-out of this gene in microglia causes a prolongation of the inflammatory response 4 and 24 h post LPS injection [175]. Interestingly, there is a possibility that this modulation is in fact mediated by OT. Namely, OT shows a high positive correlation with serotonin levels, in that OT, provokes the release of serotonin in several brain regions of the limbic system, including the amygdala [176,177]. OT also upregulates the availability of the 5-HT1A receptor [178]. This effect is produced due to the presence of OTR expression on serotonergic cells [179]. Yet, also functionally active OTR/5-HTR_2C_ and OTR/5-HTR_2A_ heterocomplexes have been identified, further emphasizing the signaling crosstalk between OT and serotonin [180]. Although it has not been tested for that direct purpose, these data imply that the serotonergic system is another potential, indirect, pathway of effect between OT and microglial activity in neonatal neuroinflammation. However, although the work by Bechade and colleagues [175] is in support of this notion, they only show the consequences of 5-HTR_2B_ knockout on neuroinflammation, and thus the possible protective effect via increased upregulation of this receptor remains to be tested.

## 6. Conclusions and Future Perspectives

Many perinatal complications are associated with an inflammatory response in the neonatal brain, which can have a detrimental and long-lasting impact on brain development. There is much support for the anti-inflammatory effect of OT, which produces beneficial effects on a structural, functional and behavioral level in the developing brain. In this review, we have discussed the most up-to-date knowledge concerning the possible pathways of effect between OT and the inflammatory response (Figure 3). The protective effect of OT likely goes through microglia, potentially via the eIF-2a-ATF4, MAPK or NADPH pathways, and/or through astrocytes, possibly via TGF-β, MAPK signaling or the retraction of astrocyte processes from the synapse/gap-junction (Figure 3). Nevertheless, the exact molecular mechanisms underlying the association remain widely understudied. From experimental data gathered so far, more direct links are found between OT and a reduction in microglial reactivity, rather than an effect on astrocytes. In this, it is unlikely that OT affects microglia via synaptic release because microglia are not part of the tripartite synapse. Rather, if OT acts on microglia directly, it will likely be in the form of dendritic release (in the PVN) or volume transmission to microglia in distant brain regions via axonal varicosities and the CSF (Figure 3). However, it remains questionable whether microglia express OTR in vivo, and whether these receptors are already developed in the neonatal/pediatric age. Moreover, there is the possibility that the effect of OT on glial cell functioning is influenced by not only the absence or presence of inflammation but also by the degree of inflammation. A further complicating factor is the highly interactive, bidirectional relationship between astrocytes and microglia in the neuroinflammatory response. As both cell types can re- and de-activate each other, it is challenging to identify exactly where OT has its effect. More specifically, to separate the primary effects of OT from the secondary effects mediated through intercellular signaling between the two glial cell types. That said, the support for the beneficial effects of OT treatment on the protection of the inflamed neonatal brain is strong. This emphasizes the need to further investigate the underlying molecular mechanisms, which will bring viable neuroprotective treatment options for the challenged developing brain closer in reach. To realize this, the amygdala poses an attractive region of interest to investigate the underlying mechanisms between OT and glial cell reactivity, due to its importance for the OT system, its implication in perinatal complications, and its clear description of functional associations between OT and astrocytes, and OTR-expressing astrocyte subtypes.

## Figures and Tables

**Figure 1 cells-11-03899-f001:**
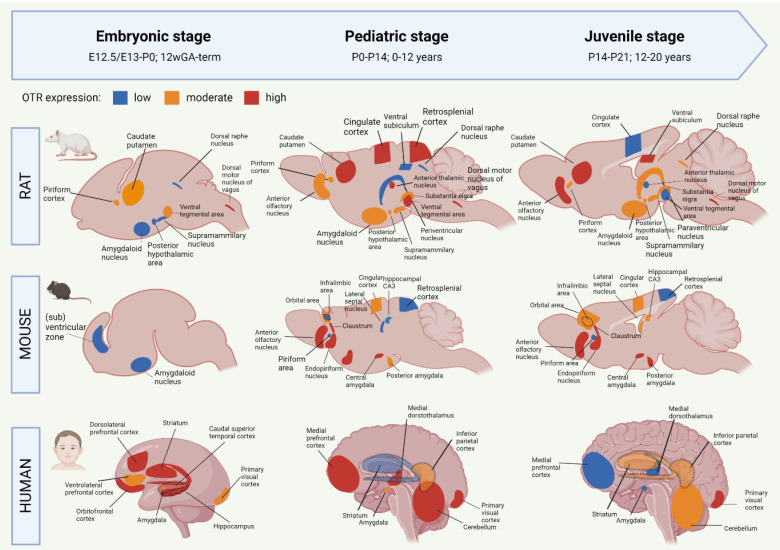
**Early life development of the oxytocin receptor system in rat, mouse and human**. Data based on OTR-binding autoradiography and OTR immunohistochemistry studies. The degree of OTR expression is visualized as low (blue), intermediate (orange) or high (red). Dashed regions that overlap other regions indicate a more lateral localization on the sagittal plane. Data based on [28,38,55,56,57,58,59]. GA = gestational age. Created with BioRender.com.

**Figure 2 cells-11-03899-f002:**
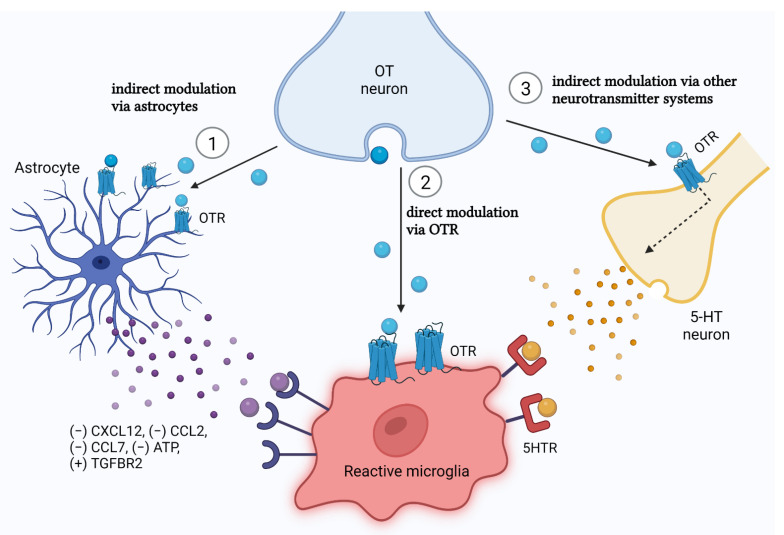
**Possible activation pathways between OT and microglial cells in neuroinflammation.** (1) OT can bind to astrocyte OTR, and reduce the reactivity of microglial cells indirectly by decreasing (“−”) secretion of chemokines CXCL12, CCL2, CCL7 and ATP from astrocytes, and increasing (“+”) activation of microglial TGFB2 receptor. (2) OT could activate microglia directly through OTR expression on microglia, yet, the presence of microglial OTR in vivo in the neonatal age is debated. (3) A third possible pathway of effect involves additional neurotransmitter systems such as serotonin (5-HT). OT can influence serotonin due to the presence of OTR on serotonergic neurons. Serotonin can bind to microglia via microglial 5-HTR expression. Created with BioRender.com.

**Figure 3 cells-11-03899-f003:**
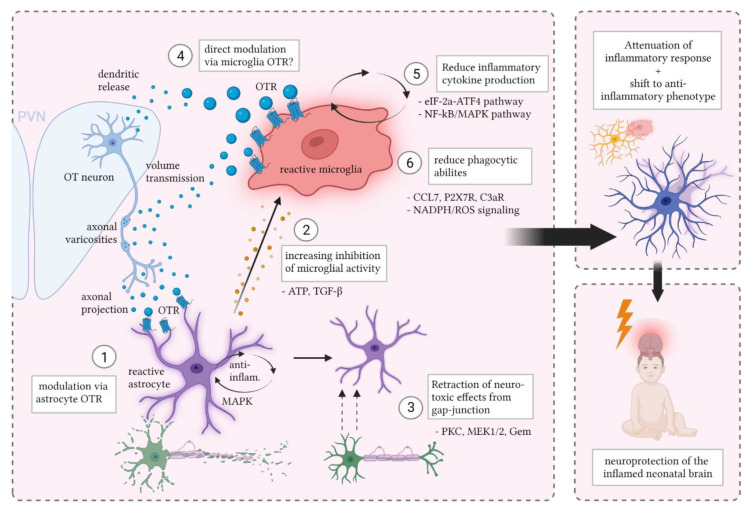
**Potential mechanisms underlying the protective effect of neonatal oxytocin treatment in neuroinflammation.** In a state of neonatal inflammation, (1) OT neurons from the PVN project OT that can bind to OTRs on reactive astrocytes. This instigates an increase in anti-inflammatory cytokine production, possibly influenced by MAPK. As a consequence, (2) astrocytes secrete signaling molecules (e.g., TGF-β and ATP) that inhibit microglial reactivity, and (3) they also possibly retract their processes from the synapse/gap-junction, thereby decreasing the neurotoxic effect. The effect of OT reaches microglia either (2) via astrocytes or (4) directly via microglia OTR expression following somatodendritic release (that reaches microglia in the PVN) or volume transmission of OT from axonal varicosities (reaching microglial cells in distant brain regions). Upon stimulation by OT, (5) microglia reduce the production of pro-inflammatory cytokines, which could involve the eIF-2a-ATF4 and NF-кB/MAPK pathways, (6) and microglia reduce their phagocytic properties, possibly via specific receptor activation, or NADPH/ROS signaling. These OT-mediated processes cause a phenotypic shift from pro- to anti-inflammatory microglia and astrocytes, which attenuates the inflammatory response, and eventually forms a neuroprotective effect during neonatal brain development. Created with BioRender.com.

**Table 1 cells-11-03899-t001:** **Overview of studies on the effect of early life oxytocin treatment in experimental models of neuroinflammation or neurodevelopmental disorders linked with neuroinflammation**. OT = Oxytocin; ASD = Autism spectrum disorders; MBP = Myelin base protein; qRT-PCR; Quantitative reverse transcription polymerase chain reaction; APC = Adenomatous Polyposis Coli; GFAP = Glial fibrillary acidic protein; aDG = Anterior dentate gyrus; TNF-α = Tumor necrosis factor alpha; IL-1β = Interleukin-1 beta.

Study; Journal	Species; Strain; Sex	Disorder Studied/Simulated + Method of Induction	Age at OT(RA) Administration	OT(RA) Administration + Concentration	Age at Readout of Effect	Readouts Assessed	Effect OT on Neuro-inflammation	Effect OT on Neuro-structural/ Functional Outcomes
[126] Glia	Rat; Sprague Dawley; both sexes	Intrauterine-growth-restriction; Low-protein diet and postnatal IL-1b injection	Twice daily administration at P1 and P2	I.p. injection of OTR agonist carbetocin (1 mg/kg)	P2; P4; P10; behavior: one to two-months old	Microarray analysis; qRT-PCR; immunohistochemistry of Iba1/MBP/APC/Olig2; functional ultrasound imaging; behavioral tests (Open field; Y-maze)	Decrease in P4 microglial activation (pro-inflammatory cytokines + gene pathways involved in inflammation).	Rescue of P10 decrease in MBP density, number of APC oligodendrocytes. No effect on total oligo number. Rescue of anxious behavior and Y-maze learning deficiencies
[122] Biol. Psychiatry	Mouse; C57BL/6J background; males only	ASD; Genetic; Magel2^+m/−p^ deficiency	Daily administration from P0-P6 First injection 3–5 h after birth	S.c. Injection 2 μg OT dissolved in 20 μL of isotonic saline	Adulthood (4 months old)	Behavioral tests: social recognition and social interaction tests, Morris water maze and Open field, Y-maze, object recognition task	Not reported	Restoration of normal quantity OT immunoreactivity in the medial amygdala. Restoration of deficits in social behavior and Morris water maze performance.
[124] Molec. Psychiatry	Mouse; C57BL/6J background; males only	ASD; Genetic; Magel2^tm1.1Mus^ deficiency	P0,P2,P4,P6	s.c. administration of 2 μg OT (1 mg/kg)	Adulthood	RNAscope (ISH) in hippocampus (aDG, CA2/CA3) Behavior: social behavior, locomotor and vertical activity, anxiety-like behavior and nonsocial memory	Not reported	Rescue of social memory deficit. No effect on sociability and social discrimination. Females: rescue Normalization of increased OT-binding sites in aDG, and increased SST+ neurons in aCA2/CA3d and aDG regions. Normalization of increased GABAergic activity in aCA2/CA3d neurons. Rescue of delayed excitatory-to-inhibitory GABA shift in hippocampal neurons.
[132] Biomed Pharmacother.	Mouse; C57BL/6J background; males only	ASD; Valproate acid treatment of pregnant dam	P30 (juvenile age)	Intranasal OT administration (200 μg/kg) 30 min before behavioral testing	P30	RT-qPCR; immunohistochemistry; behavioral assays.	Normalization of increased TNF-α, IL-1β and IL-6 expression, and attenuation of increased oxidative stress response in amygdala and hippocampus.	Rescue of autism-like behavior in the open field, tail suspension test, social interaction test and marble burying test.
[130] Peptides	Rat; Sprague Dawley; both sexes	neonatal hypercapnic-hypoxia injury; P0 exposure to 100% CO_2_ (5 or 10 min) At 6 months: pentylenetetrazol-induced seizures	P1–P28 (1 month)	I.P., 100 IU/kg/day oxytocin	6 months	GFAP Immunohistochemistry in CA1, ELISA (TNF-α + GAD-67)	Rescue of increased TNF-α levels, and of decreased GAD-67 levels. Reduction in astrogliosis (number of GFAP+ cells in CA1)	Not reported

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
