# Peer review of "The Role of Oxytocin in Abnormal Brain Development: Effect on Glial Cells and Neuroinflammation"

_cells, 2022, doi:10.3390/cells11233899_

Round 1
Reviewer 1 Report
The work submitted for review provides a very comprehensive and well-structured overview of the oxytocinergic system in the CNS in the context of normal ontogeny as well as in consequence of perinatal complications. The effects of spatiotemporal impaired oxytocin (OT) and oxytocin receptor (OTR) synthesis on anatomical and social pathologies at the animal level as well as in the human system are extensively discussed. The focus of possible triggers on OT system-dependent developmental disorders is on neuroinflammatory processes. Their understanding is rightly seen as essential for the development of promising therapeutic approaches.
The quality of the manuscript is very high, the literature cited is current and of a high linguistic standard. I have only a few comments that I believe should be considered in the manuscript. However, based on the high quality of the manuscript, their implementation is rather obligatory.
The criticisms in detail:
In Figure 1, the cartoon for the human brain at the embryonic stage is decidedly too small and in no way contains information on spatial expression as is beautifully shown in all other schematic sagiall sections. Here, a detailed cartoon should be chosen on which also the area of the corpus amygdaloideum is sketched.
Since the corpus amygdaloideum plays an essential role as a reference nucleus complex in the manuscript, information on its location and structure should be briefly mentioned. A possible citation here would be: van Staalduinen EK, Zeineh MM. Medial Temporal Lobe Anatomy. Neuroimaging Clin N Am. 2022 Aug;32(3):475-489. doi: 10.1016/j.nic.2022.04.012. PMID: 35843657.https://www.sciencedirect.com/science/article/pii/S1052514922000387?via%3Dihub#fig11
Line 324: The periaqueductal and central gray are also directly involved in the descending pain inhibitory system. This information can be found later in the text, but should also be mentioned here.
Line 382: The authors should specify more precisely what they mean by "disease". Do you mean bacterial / viral inflammations, tumor diseases or mutations (e.g. chromosomal like trisomy 21)?
Line 424-426: A meta-analysis of 424 studies comprising 1422 participants found lower levels of endogenous OT in the blood 425 plasma or saliva of children with ASD compared to neurotypical controls (Moerkerke et al., 2021) This information is used semantically as a finding for the statement that OT is an important player in brain development at both the functional and structural level. However, there is the unanswered question of how OT, which is systemically present in the blood and thus more likely to be regarded as a hormone, can have an effect on neurogenesis. The cited study discusses exactly this point (page 12-13). For the sake of correctness, this information should also be included in the manuscript.
Line 454: How can you say that the activity of OT decreased? You also give information of spatiotemporal OTR expression. Reduced OTR level might also lead to decreased OT dependent stimulation without reducing the OT activity by itself.
Line 484: Where are the pro-inflammatory factors IL-6, IL-1β, TNF-α, and iNOS elevated? In Blood or CSF?
Line 660: Please specify the BV-2 cells as murine microglial cell line when correct.
Line 712: … OT treatment during inflammatory states affects the number of astrocytes, … DO the astrocytes proliferate or do you mean the number of GFAP-positive glial cells increase by enhanced GFAP expression in single cells?
Line 742-743: How can you speak of a healthy brain milieu when the analyses were carried out in vitro on primary cells?
Line 733: Here you talk about microglia phenotypes. Which ones do you mean, the pro-inflammatory M1 and the anti-inflammatory M2 state or based on ramification? Overall, it would be good to briefly discuss these functionally different microglia phenotypes.
Author Response
We sincerely thank reviewer 1 for her/his time to read the review and the constructive criticism provided. We have changed the manuscript in line with those comments. Please, find below responses to each individual comment, illustrating the changes that were applied to the manuscript (or, in a few cases, explaining why no changes were made).
Best regards,
Knoop and O. Baud, (first and last authors).
Reviewer 1:
The criticisms in detail:
In Figure 1, the cartoon for the human brain at the embryonic stage is decidedly too small and in no way contains information on spatial expression as is beautifully shown in all other schematic sagiall sections. Here, a detailed cartoon should be chosen on which also the area of the corpus amygdaloideum is sketched. Since the corpus amygdaloideum plays an essential role as a reference nucleus complex in the manuscript, information on its location and structure should be briefly mentioned. A possible citation here would be: van Staalduinen EK, Zeineh MM. Medial Temporal Lobe Anatomy. Neuroimaging Clin N Am. 2022 Aug;32(3):475-489. doi: 10.1016/j.nic.2022.04.012. PMID: 35843657.https://www.sciencedirect.com/science/article/pii/S1052514922000387?via%3Dihub#fig11
Reply from authors: Thank you for the remark. We changed figure 1 so that it now shows a spatial expression similar to the other sagittal sections, including the amygdala.
Line 324: The periaqueductal and central gray are also directly involved in the descending pain inhibitory system. This information can be found later in the text, but should also be mentioned here.
Reply from authors: Thank you for the remark. We have inserted this piece of information into line 324.
Line 382: The authors should specify more precisely what they mean by "disease". Do you mean bacterial / viral inflammations, tumor diseases or mutations (e.g. chromosomal like trisomy 21)?
Reply from authors: Thank you for bringing this to our attention. We have clarified in the manuscript that it refers to bacterial and viral infections.
Line 424-426: A meta-analysis of 424 studies comprising 1422 participants found lower levels of endogenous OT in the blood 425 plasma or saliva of children with ASD compared to neurotypical controls (Moerkerke et al., 2021). This information is used semantically as a finding for the statement that OT is an important player in brain development at both the functional and structural level. However, there is the unanswered question of how OT, which is systemically present in the blood and thus more likely to be regarded as a hormone, can have an effect on neurogenesis. The cited study discusses exactly this point (page 12-13). For the sake of correctness, this information should also be included in the manuscript.
Reply from authors: Thank you for mentioning this. We fully agree and have added a nuancing statement to the manuscript, stating that peripheral OT cannot give any conclusions about central OT activity.
Line 454: How can you say that the activity of OT decreased? You also give information of spatiotemporal OTR expression. Reduced OTR level might also lead to decreased OT dependent stimulation without reducing the OT activity by itself.
Reply from authors: The statement that OT activity is decreased, refers to the findings from Mezianet et al., 2015, Schaller et al., 2010 and Dai et al., 2018, who showed a decrease in number of OT-positive neurons as well as a lower concentration of (mature) OT in the cerebrum and cerebral spinal fluid in neonatal pups from rodent models of ASD. Also, Mairesse et al., 2019 showed decreased levels of hypothalamic OT in neonatal rat pups that were injected with pro-inflammatory cytokines. In this section we only talk about OT neurons and OT peptide concentrations. We do not discuss OTR expression. Although the reviewer is correct in saying that a decrease in OTR expression would affect OT activity without changing the level of OT peptide by itself, we believe this is not applicable to the highlighted section because we do not mention any changes in the OTR system as a consequence of neuroinflammation and related neurodevelopmental disorders.
Line 484: Where are the pro-inflammatory factors IL-6, IL-1β, TNF-α, and iNOS elevated? In Blood or CSF?
Reply from authors: Thank you for bringing this to our attention, we have clarified that it concerns pro-inflammatory markers in the cortex (secreted by cortical microglia).
Line 660: Please specify the BV-2 cells as murine microglial cell line when correct.
Reply from authors: Thank you for the remark, we have replaced “BV-2 cells” with “BV-2 murine microglial cell line”.
Line 712: … OT treatment during inflammatory states affects the number of astrocytes … DO the astrocytes proliferate or do you mean the number of GFAP-positive glial cells increase by enhanced GFAP expression in single cells?
Reply from authors: Thank you for bringing this discrepancy to our attention. The line referred to an increase in GFAP+ cell number, thus astrocyte proliferation, rather than an increase in GFAP-signal within the cells. We have changed the manuscript accordingly.
Line 742-743: How can you speak of a healthy brain milieu when the analyses were carried out in vitro on primary cells?
Reply from authors: Thank you for the remark. We fully agree and have replaced ‘healthy brain milieu’ by ‘baseline conditions’.
Line 733: Here you talk about microglia phenotypes. Which ones do you mean, the pro-inflammatory M1 and the anti-inflammatory M2 state or based on ramification? Overall, it would be good to briefly discuss these functionally different microglia phenotypes.
Reply from authors: We thank the reviewer for this remark. We understand the urge for a M1/M2 distinction, however a recent publication from a ‘microglia collective/consortium’ pleads to refrain from further use of the M1/M2 classification because it is too generalizing and misleading (see Paolicelli et al., 2022: Microglia states and nomenclature: A field at its crossroads. Neuron, 110(21), 3458–3483. https://doi.org/10.1016/j.neuron.2022.10.020). We, therefore, prefer not to include the M1/M2 terminology in our manuscript. However, we have added a few sentences in the introduction to address the M1/M2 terminology. And at Line 733 we have added ‘reactive’ to emphasize that it concerns the silencing control of astrocytes on reactive/pro-inflammatory microglia.
Reviewer 2 Report
The review is comprehensive, deep, and broad. I found the manuscript is well written and the data and results support their conclusions. I have some minor suggestions for authors to consider.
Minor comments and suggestions:
1. Shorten the description of basic neuroinflammation mechanisms.
2. Table 1 should be visible clearly.
Author Response
We sincerely thank reviewer 2 for her/his time to read the review and the constructive criticisms provided. We have provided below a response to each individual comment, illustrating the changes that were applied to the manuscript (or, in a few cases, explaining why no changes were made).
Bestregards,
Knoop and O. Baud, (first and last authors).
Reviewer 2:
Minor comments and suggestions:
- Shorten the description of basic neuroinflammation mechanisms.
Reply from authors: Thank you for the feedback. However, we feel that, in order to give a complete story, to make the review understandable for readers from multiple backgrounds, and to give the reader sufficient understanding of neuroinflammation so that he/she/they can understand the in-depth section 5, a few sentences should be spent on the introduction of neuroinflammation and its basic workings. We feel that a further reduction would go at the cost of the comprehension of the review.
- Table 1 should be visible clearly.
Reply from authors: Thank you for this suggestion. As this is an editing matter, we as authors do not know exactly how to improve this. We look forward to receiving feedback from the editors about how to improve the visibility of Table 1.